# Monitoring Systems in Home Ventilation

**DOI:** 10.3390/jcm12062163

**Published:** 2023-03-10

**Authors:** Jean-Michel Arnal, Mathilde Oranger, Jésus Gonzalez-Bermejo

**Affiliations:** 1Service de Réanimation Polyvalente et Unité de Ventilation à Domicile, Hôpital Sainte Musse, 83100 Toulon, France; 2Service de Pneumologie, Médecine Intensive et Réanimation (Département R3S), Groupe Hospitalier Universitaire APHP-Sorbonne Université, Site Pitié-Salpêtrière, 75013 Paris, France; 3INSERM, UMRS1158 Neurophysiologie Respiratoire Expérimentale et Clinique, Sorbonne Université, 75006 Paris, France

**Keywords:** noninvasive ventilation, chronic respiratory failure, monitoring, telemonitoring

## Abstract

Non-invasive ventilation (NIV) is commonly used at home for patient with nocturnal hypoventilation caused by a chronic respiratory failure. Monitoring NIV is required to optimize the ventilator settings when the lung condition changes over time, and to detect common problems such as unintentional leaks, upper airway obstructions, and patient–ventilator asynchronies. This review describes the accuracy and limitations of the data recorded by the ventilator. To efficiently interpret this huge amount of data, clinician assess the daily use and regularity of NIV utilization, the unintentional leaks and their repartition along the NIV session, the apnea–hypopnea index and the flow waveform, and the patient–ventilator synchrony. Nocturnal recordings of gas exchanges are also required to detect nocturnal alveolar hypoventilation. This review describes the indication, validity criteria, and interpretation of nocturnal oximetry and transcutaneous capnography. Polygraphy and polysomnography are indicated in specific cases to characterize upper airway obstruction. Telemonitoring of the ventilator is a useful tool that should be integrated in the monitoring strategy. The technical solution, information, and limitations are discussed. In conclusion, a basic monitoring package is recommended for all patients complemented by advanced monitoring for specific cases.

## 1. Introduction

Non-invasive ventilation (NIV) is effective for the correction of nocturnal hypoventilation and prevention of upper airway obstructions that occur during sleep in patients with chronic hypercapnic respiratory failure. In many different types of chronic respiratory failure resulting from chronic obstructive respiratory disease (COPD) and other obstructive lung diseases, in obesity hypoventilation syndrome and overlap syndrome, in various restrictive thoracic disorders, and in neuromuscular diseases, NIV improves sleep quality and quality of life, decreases the number of exacerbations and hospital admissions, and may improve survival [1]. NIV requires regular monitoring to optimize the ventilator settings during the initiation period and to readjust them when the lung condition changes over time. In addition, problems such as unintentional leaks, upper airway obstructions, and patient–ventilator asynchrony may occur and impair treatment quality [2]. The overall efficacy of the treatment is assessed by monitoring the clinical symptoms of nocturnal hypoventilation, as well as the daytime measurement of blood gas exchanges. However, a precise assessment of the quality of NIV used at night requires more detailed monitoring. This review looks at the monitoring of NIV used at home for an adult population, with a focus on the respective values and limitations of data recorded by the ventilator, the nocturnal measurement of gas exchanges and patient effort, and the possibilities offered by telemonitoring solutions.

## 2. Data Recorded by the Ventilator

Modern ventilators record all ventilation variables at a high sample rate, providing a reliable source of information to assess the quality of NIV. Recordings are stored on a memory card or can be downloaded to a USB stick, and can be interpreted using a manufacturer-specific software. Results are provided in form of statistics, trends, and waveforms. In order to interpret these data correctly, the clinician should be aware of how precise the measurement and calculation for each variable is (Table 1).

### 2.1. Measurements and Calculations

Pressure and flow are measured by a pneumotachograph positioned at the ventilator’s output. From these measurements, the airway pressure, patient’s flow, and leaks are calculated, displayed on the ventilator screen, and recorded.

#### 2.1.1. Airway Pressure

Airway pressure is calculated as ventilator output pressure minus the pressure lost due to resistance in the ventilator circuit. Airway pressure is more accurate when the ventilator circuit diameter is correctly set or when the circuit is calibrated. Airway pressure accuracy is ±0.5 cm H_2_O, which is considered to be a reliable measurement clinically.

#### 2.1.2. Flow

Patient flow is calculated from the total flow measured at the ventilator output minus the mean leaks. Patient flow accuracy is ±10%, which is mainly due to the characteristics of the pneumotachograph used in home-care ventilators. However, this accuracy is clinically acceptable.

#### 2.1.3. Volume

The tidal volume and, by extrapolation, the minute volume are calculated from the patient’s flow. Manufacturers claim that the precision of volume is ±20%. Again, this is due to the characteristics of the pneumotachograph and the inability of the ventilator to measure the exact temperature and humidity of the inspired gas. In addition, unintentional leaks affect the estimate of tidal volume [3]. Therefore, the tidal volume provided by home-care ventilators should be interpreted with caution.

#### 2.1.4. Leaks

Leaks are calculated from the total flow and presented either as total or unintentional leaks. While the calculation of total leaks is accurate, that of unintentional leaks is less accurate because the ventilator is not able to measure the exact intentional leak [4].

#### 2.1.5. Other Measurements

All measurements of time such as inspiratory and expiratory time, the I:E ratio, and the percentage of spontaneous triggering and cycling are accurate. The respiratory rate should be interpreted as the ventilatory rate. The ventilator measures the number of mechanical breaths. Depending on the degree of patient–ventilator synchrony, the patient’s respiratory rate can be very different from the ventilatory rate.

### 2.2. Interpretation

Ventilators record a huge amount of data. In order to be efficient, the clinician should use a standardized method of interpretation: 1—the daily use of the treatment, 2—the leaks, 3—the upper airway obstructions, 4—patient–ventilator synchrony.

#### 2.2.1. Daily Use

Daily use of the treatment is an important piece of information because NIV is efficient if used for at least 5 h per day during sleep [5,6]. The median value of daily use is preferred to the mean value, which is affected by extreme values. In addition to median daily use, the pattern of use is also very important. The clinician checks that the patient uses NIV at night on a daily basis and preferably at regular hours, and then determines the number of NIV sessions. Sub-optimal median daily use, irregular use, or multiple sessions are usually signals that common problems such as leaks, upper airway obstructions, or patient–ventilator asynchronies are present.

#### 2.2.2. Leaks

Monitoring leaks is very important, but it is also complex because manufacturers measure and display different metrics for leaks. In general, manufacturers show total leaks or unintentional leaks. Total leaks are more accurate, but the mean or median total leak is difficult to interpret because the intentional leak is difficult to determine. In contrast, unintentional leaks are less accurate because the ventilator does not measure the intentional leaks; however, they are easier to interpret. Different clinicians use different thresholds to interpret statistics for unintentional leaks. Median unintentional leaks below 10–12 L/min and 95th percentile below 20–24 L/min are commonly accepted. Median unintentional leaks are calculated over the full breath. They are accurate when leaks occur both at inspiration and expiration. However, they are underestimated when leaks occur during inspiration over a certain pressure. Very large unintentional leaks above 40–50 L/min should be avoided, because ventilators are not able to maintain inspiratory pressure above this threshold. The trend of median leaks over time provides information about the deterioration of the interface. However, to determine the cause of unintentional leaks, the clinician may interpret the repartition of leaks throughout the night. There are several typical profiles of leak repartition overnight (Figure 1).

#### 2.2.3. Upper Airway Obstructions

Upper airway obstructions are common in home-care NIV with different localization and mechanisms. Most manufacturers use an algorithm based on the decrease in peak inspiratory flow over a certain period of time to automatically detect upper airway obstructions and provide an apnea–hypopnea index (AHI). AHI has acceptable accuracy for stable patients with obesity hypoventilation syndrome in the absence of unintentional leaks [7,8]. However, AHI fails to detect upper airway obstructions in the case of unintentional leaks and laryngeal obstructions. Therefore, an examination of the flow waveform may help to detect some events not included in the AHI (Figure 2). In the case of upper airway obstruction, the shape of the flow waveform may give an indication as to the mechanism of obstruction. However, a polygraphy is required to confirm it.

#### 2.2.4. Patient–Ventilator Synchrony

Patient–ventilator synchrony is assessed by the statistics and can be analyzed in detail on the pressure and flow waveforms. The overall patient–ventilator synchrony is shown together with the percentage of spontaneously triggered and cycled breaths. These two variables are dependent on many others and should be interpreted with caution. The percentage of triggered breaths depends on the setting of the inspiratory trigger and respiratory rate. It is overestimated in the case of auto-triggering. Conversely, it does not take into account the presence of ineffective efforts and can also be affected by unintentional leaks. The percentage of spontaneously cycled breaths depends on the settings for cycling criteria, as well as the minimum and maximum inspiratory time. In addition, it is also affected by unintentional leaks. A detailed analysis of patient–ventilator synchrony is made by interpreting the pressure and flow waveform over a focused timescale axis such as 1 min per epoch [9,10]. Phase asynchronies such as ineffective efforts, auto-triggering, premature cycling, double triggering, and delayed cycling can be recognized. In addition, flow asynchronies such as flow overshoot or insufficient flow can also be identified (Figure 3).

#### 2.2.5. Other Variables

Other variables are recorded by the ventilator. Tidal volume and minute volume are not of great interest clinically because their accuracy is sub-optimal. However, in the absence of unintentional leaks, a low tidal volume can be considered correct and should be taken into account. The ventilatory rate is dependent on settings, leaks, and patient–ventilator synchrony. Inspiratory time can be monitored to check that the minimum and maximum inspiratory times are set correctly.

Data recorded by the ventilator can be interpreted directly on a computer using the manufacturer-specific software or on a report generated by this software. The number of variables and how they are displayed on a report vary considerably according to the manufacturer. In the best-case scenario, it may be possible to customize the report with all the desired information regarding daily use and leaks. However, only those statistics for upper airway obstructions and patient–ventilator synchrony are available on a report. A summary of data recorded by the ventilator can be monitored remotely using telemonitoring.

## 3. Nocturnal Recordings

Nocturnal hypoventilation occurs mainly during the rapid eye movement (REM) stage of sleep in patients with impaired lung function and affects the nocturnal exchange of gas with concomitant hypoxemia and hypercapnia. The overall efficacy of NIV cannot be assessed by measuring the daytime gas exchange only, because around 25% of patients with normal morning or daytime gas exchanges present with persistent nocturnal hypoventilation [11,12]. Therefore, nocturnal gas exchanges need to be assessed by nocturnal pulse oximetry and/or by transcutaneous capnography.

### 3.1. Nocturnal Pulse Oximetry

Nocturnal pulse oximetry (SpO_2_) is a low-cost, non-invasive tool that continuously measures the arterial blood oxygen saturation (SaO_2_) with a bias of less than 2% and precision of less than 3% if SaO_2_ is above 90% [13]. The response time of the conventional pulse oximeters using transmission sensors is short, which makes the measurement suitable for capturing the rapid and short desaturation that occurs during upper airway obstructions. Because it is simple to use, there are very few failures of non-supervised recording at home, provided that the size of the finger is suitable for the patient and it is secured to prevent motion artifacts. Nocturnal pulse oximetry is not reliable in the case of dyshemoglobins such as methemoglobin, carboxyhemoglobin, and homozygous sickle cell disease. Erroneous measurements can also result from a low-perfusion state, anemia, and nail polish [13]. However, the main limitation to using overnight recording is the shape of the hemoglobin dissociation curve. Patients with SaO_2_ above 96% at baseline will only show a decrease in SpO_2_ if there is a very large decrease in the arterial pressure of oxygen (PaO_2_). As a consequence, respiratory events defined by a specific change in saturation are less likely to be detected, which means decreased sensitivity [14]. Therefore, nocturnal pulse oximetry is not indicated in pediatric and young adults with neuromuscular disease if they present with normal lung function and baseline SaO_2_ above 95%. An important technical aspect comes from the signal processing, namely the sample frequency and the averaging time. Amongst the sensors currently commercially available, there is a large discrepancy in frequency ranging from one recording per second to one recording every 12 s. This setting affects the results provided by the statistics and the ease of visually inspecting the curves. Beside these limitations, nocturnal pulse oximetry can be used to screen for nocturnal respiratory events [15], to predict the outcome [16,17], and to monitor patients using NIV at night. Sensibility and specificity are not optimal when pre-test probability of having a specific disease is low. Therefore, clinicians should use it with caution as screening tool. Conversely, in patients using NIV at night, the pre-test probability of having upper airway obstruction and/or nocturnal hypoventilation is high, which makes nocturnal pulse oximetry a valuable tool. Nocturnal pulse oximetry can be interpreted when the recording duration is at least 4.5 h of sleep with less than 20% of artifacts, and when the information about sleeping times and unintentional leaks is available. Because unintentional leaks cause desaturation, period of unintentional leaks should be discarded from the recording to prevent misinterpretation. Nocturnal pulse oximetry is interpreted by looking at the statistics provided automatically by the manufacturer. Currently accepted threshold values come from studies primarily including patients with neuromuscular disease that do not stipulate the sampling frequency and that are calculated based on the full recording without discarding the awake periods. However, mean SpO_2_ below 90%, oxygen desaturation index (ODI) at 4% above 5 or 10 per hour, and time spent with SpO_2_ below 90% are considered as abnormal results [18]. Due to all the limitations surrounding statistics, the visual inspection of traces is very informative. In the absence of unintentional leaks, the desaturation shape and duration suggest upper airway obstruction, nocturnal hypoventilation, or ventilation/perfusion mismatch [19] (Figure 4). Sleep fragmentation can be assessed by pulse wave amplitude [20]. While a normal nocturnal pulse oximetry recording can be associated with persistent abnormality on transcutaneous capnography [21,22], an abnormal nocturnal pulse oximetry recording should result in adjustments to the ventilator settings or interface. Therefore, nocturnal pulse oximetry can be used as a simple, low-cost screening tool to assess the overall quality of NIV on gas exchanges.

Nocturnal pulse oximetry can be performed using a stand-alone device or a sensor connected to the ventilator. The latter enables a visual examination of the SpO_2_ trace together with unintentional leaks, and the pressure and flow waveforms (Figure 5). However, the sampling frequency is usually lower, and the statistics are less comprehensive, with no possibility to discard any measurements. Currently, telemonitoring of nocturnal pulse oximetry is possible, but the sampling frequency is not appropriate for monitoring NIV.

### 3.2. Transcutaneous Capnography

Transcutaneous capnography uses a skin electrode that heats the local capillaries in order to arterialize the local blood circulation. CO_2_ diffuses via the tissues and is measured continuously by the skin electrode [23]. Therefore, PtcCO_2_ is a surrogate for PaCO_2_. New-generation sensors combine PtcCO_2_ and SpO_2_ measurements. The electrode needs baseline calibration before use and also takes some time to stabilize; in addition, electronic drift occurs over time and makes a second calibration necessary at the end of the recording to correct the measurement retrospectively. Incorrect positioning of the sensor, trapped air bubbles between the skin and the sensor, incorrect calibration of the sensor, failure to correct the drift, and damage to the sensor membrane are the technical causes of measurement failure [24]. Due to these causes, transcutaneous capnography fails more often when measured at home without supervision than it does in an inpatient setting [25,26]. There is a good correlation between PtcCO_2_ and PaCO_2_ with a bias of less than 5 mmHg, but the limits of agreement are greater than 7 mmHg in most of the studies [27,28]. Because the limits of agreement are in the same range as the expected effect of NIV on PaCO_2_, using raw measurements or statistics based on raw measurements is questionable. In addition, there is a response time of 2 min when PaCO_2_ is changing. Therefore, transcutaneous capnography is a surrogate for the trend of PaCO_2_ during the night. A nocturnal recording of transcutaneous capnography can be used to confirm nocturnal hypoventilation in spontaneously breathing patients or patients using CPAP to indicate nocturnal NIV [12,28,29]. In patients using NIV, transcutaneous capnography is used to monitor the quality of the treatment [22,30]. Nocturnal transcutaneous capnography is considered valuable when the recording duration is at least 4.5 h of sleep with less than 20% of artifacts, when PtcCO_2_ is between 30 and 70 mmHg with an individual variation of less than 20 mmHg during a single night, and when the information about sleeping times and unintentional leaks is available. Unintentional leaks can lead to a transient increase in PtcCO_2_ that should not be interpreted as nocturnal hypoventilation. Nocturnal hypoventilation is diagnosed based on the statistics for PtcCO_2_ provided automatically by the manufacturer. However, there is no agreement regarding the definition of nocturnal hypoventilation, and different definitions lead to different results [12]. Some authors use raw values such as maximum PtcCO_2_ > 49 mmHg [21], or mean PtcCO_2_ > 50 mmHg [11,22], while others use a change in PtcCO_2_ from baseline to a maximum value above 7.5 or 10 mmHg [25,29]. However, the optimal clinical threshold for defining nocturnal hypoventilation may differ according to the device used, the method of use (e.g., calibration or drift correction), the clinical condition (COPD, SOH, or neuromuscular disorders), the baseline PtcCO_2_, and the clinical setting (patient in spontaneous breathing or on NIV) [2,22]. Due to all the limitations surrounding statistics, a visual inspection of the changes in PtcCO_2_ during the night may better indicate nocturnal hypoventilation or hyperventilation (Figure 6).

Transcutaneous capnography is usually measured by a stand-alone device. It is now possible to connect the ventilator and the capnograph so the curve of leaks, flow, tidal volume, and PtcCO_2_ can be displayed on the same screen (Figure 7). This combination may help us to understand the mechanism of abnormal respiratory events of uncertain classification, such as ventilatory over-assistance causing upper airway obstructions due to a decrease in respiratory drive [31].

### 3.3. Respiratory Polygraphy

Respiratory polygraphy adds an assessment of the patient’s effort to the flow and airway pressure using a thoracic and an abdominal belt. When the upper airways are open, the signal from the thoracic and abdominal belts increases during the mechanical breath for both the patient’s triggered and controlled breaths. However, when the upper airways have collapsed, the belts provide information about the absence or presence of patient effort and thus help characterize the mechanism of the obstruction. Upper airway obstruction without patient effort suggests laryngeal obstruction with a decreased respiratory drive. Upper airway obstruction with patient effort is characterized by paradoxical thoracoabdominal movements that occur in case of pharyngeal collapse (Figure 8). The use of polygraphy in patients on NIV has been described in detail by the SomnoVNI group [32]. Unattended polygraphy is feasible at home for patients using NIV with fewer recording failures than transcutaneous capnography [25]. However, interpretation of the recording remains manual and requires time and expertise. While some specialized centers use polygraphy for titration and follow-up, most authors recommend performing polygraphy in selected patients [2]. Upper airway obstructions are detected by the inbuilt ventilator software with relative accuracy for stable patients with obesity hypoventilation syndrome in the absence of unintentional leaks [7,8]. However, inbuilt ventilator software fails to detect upper airway obstructions in the case of unintentional leaks and laryngeal obstructions. Therefore, polygraphy is mainly used to characterize the mechanism of upper airway obstructions. It can also be used to analyze complex patient–ventilator asynchronies when the reading of detailed pressure and flow waveforms is not conclusive. On rare occasions, polygraphy may demonstrate paradoxical thoracoabdominal movements occurring with open upper airways, which indicate a weakness of the diaphragm.

Respiratory polygraphy can be performed with a stand-alone device that combines a proximal flow sensor inserted onto the ventilator circuit, SpO_2_ measurement, and belts. Modern ventilators are available with modules to connect the belt and SpO_2_ directly to the ventilator. This solution makes recording easier, and the clinician can see the airway pressure and flow measured by the ventilator, the effort belts, and SpO_2_ all displayed on the same page [33] (Figure 9).

### 3.4. Polysomnography

Improving sleep quality is one of the major objectives of NIV used at home overnight [34]. Polysomnography assesses both the efficiency and architecture of sleep, and may provide information on the occurrence of specific respiratory events at different sleep stages. However, full polysomnography in a patient treated with NIV is a complex and costly means of monitoring. While it is commonly used in pediatric NIV, it is seldom used in adult populations outside of clinical research and specialized centers [35]. In adult patients with neuromuscular disorders, NIV titrated during the daytime followed by polysomnography was associated with fewer patient–ventilator asynchronies than NIV titrated during the daytime alone, but the sleep quality was similar for both [36].

## 4. Telemonitoring

Over the last 20 years, technological advances have enabled the emergence of telemedicine and remote medical monitoring in several areas, with the aim of transmitting patient information directly to healthcare professionals. This could be a particularly interesting option for monitoring patients with chronic diseases at risk of exacerbation. Remote monitoring improves access to care and could impact positively on both the continuity of care and the reduction in health costs by reducing emergency room visits and hospitalizations. Moreover, the recent integration of remote continuous positive airway pressure (CPAP) monitoring in the management of obstructive sleep apnea syndrome has improved patient compliance [37] and quality of life [38], allowing earlier technical intervention at home [39]. Therefore, telemonitoring could significantly change the management of patients with chronic respiratory failure treated with long-term NIV.

### 4.1. Ventilator Data Collection Methods

The first technical solution was launched in 2014 and there have since been several advancements, including an increase in the number of variables monitored and the sample rate, and an improved display. Today, almost all home-care ventilators offer a telemonitoring solution with large variation between manufacturers in terms of usability and variables monitored [40]. Ventilator variables available on cloud-based platforms are less precise, and the frequency of sampling is lower when compared to data downloaded manually from the ventilator. However, the clinician has remote access to data on current and past usage, as well as performance. Some ventilators have inbuilt connectivity, while others require connection to an external modem or use a Bluetooth cellular hub. Data are encrypted and sent at least once per day to a server that can be accessed by a password-protected account for each clinician or home-care provider. Ventilator manufacturers provide proprietary software to access to the data.

### 4.2. Variables Telemonitored

Telemonitoring provides information about daily use, leaks, the upper airway obstruction index, tidal volume, minute volume, ventilatory rate, and a global index of triggering and cycling synchronization (Figure 10). Detailed breath-by-breath data are not available with telemonitoring, except for one manufacturer. Some manufacturer offers telemonitoring for nocturnal oximetry, but the sampling rate is only one measurement per minute. Therefore, it is not possible to detect rapid events such as desaturations caused by upper airway obstructions.

Daily compliance is the cornerstone of treatment and is an easy parameter to monitor remotely. The total daily use of NIV is provided by all manufacturers in the form of statistics and trends. In addition, some manufacturers show the detailed daily use including the time of use and the number of sessions. Decreased use of ventilation may reflect poor tolerance of the treatment in relation to adverse effects or inadequate settings [41]. In addition, low daily use was associated with lower survival rates [42].

Unintentional leaks in NIV are a frequent adverse event and have an impact on the ventilation quality. They lead to a decrease in compliance, the occurrence of patient–ventilator asynchrony, and a deterioration in sleep quality. The estimation of unintentional leaks seems to be relatively reliable for several home ventilators [43]. Furthermore, the presence of total leakage less than 30 l/min seems to be compensated for by most algorithms [44]. While the detection of intentional and unintentional leaks seems to be reliable, their presence significantly impacts the accuracy of the evaluation of other data. For example, they lead to an inaccurate estimate of tidal volume, which may be underestimated or overestimated throughout the respiratory cycle [45,46]. Telemonitoring of leaks is a major feature that helps clinician and home-care providers to manage the leaks remotely. Leaks are displayed in the form of statistics and trends, and some manufacturers display the leak curve overnight with a one-minute sample rate. This information helps determine the mechanism of the leakage and enable leak resolution, which is associated with improved NIV efficiency [47]. The use of ventilator data facilitates early, objective leak detection and the subsequent adaptation of parameters [48].

The upper airway obstruction index, namely AHI, is available with telemonitoring. The estimation of the number of residual obstructive airway events in NIV may not be accurate; however, good discrimination has been reported for some home devices in patients with an AHI > 10/h [7]. The presence of upper airway obstruction during NIV was associated with poorer survival in ALS patients [16].

Other parameters may also be of interest to monitor. Daily variations in respiratory rate and the percentage of respiratory cycles triggered by the patient were predictive of exacerbation in COPD patients treated with long-term NIV [49]. Two studies confirmed that the ventilatory rate is higher and increases in the days preceding hospitalization for COPD exacerbation [50,51]. However, finding a threshold for an alert is challenging because the signal is in the same range as the normal variations of respiratory rate in stable patients [52].

### 4.3. Applications

The two main applications of telemonitoring are remote monitoring for initiation of home NIV and for patient follow-up. Recent evidence supports the initiation of NIV at home. In COPD, hypoventilation obesity syndrome, and neuromuscular diseases, there is no inferiority in terms of effectiveness in the correction of gas exchange and quality of life when compared to conventional hospital initiation. Moreover, access to care seems to be improved with earlier initiation of NIV and there appears to be a reduction in health costs [53,54,55]. In this case, telemonitoring is very useful for monitoring the patient remotely and adapting the interventions accordingly.

The management of telemonitoring for home NIV in usual care is not well defined. The most important variables such as daily use, leaks, AHI, and respiratory rate can be screened on a regular basis by clinicians or the home-care provider in order to detect when troubleshooting is needed and to take action between the planned visits. Another option is to use alerts automatically generated by the software. While several systems offer the option of setting alerts for the main variables, it is very difficult to determine individualized thresholds for alerts with an acceptable sensitivity and specificity. In future, artificial intelligence may help clinicians to define proper alert thresholds.

### 4.4. Clinical Utility and Evidence

Telemonitoring may improve the quality of NIV, but only if it prompts decisions to be made and action to be taken. The most common actions resulting from information gathered by telemonitoring are the reinforcement of therapeutic education and modification of the treatment by adjusting either the interface or ventilator settings. Remote adjustment of ventilator settings is available for non-life-support ventilators. While it is certainly useful in many instances for making small adjustments to ventilator settings, there needs to be an appropriate framework in terms of decision-making, responsibility, and legal aspects.

Evidence regarding the benefits of telemonitoring is scarce. Telemonitoring may reduce the frequency of exacerbations and hospitalizations in COPD [56,57]. In patients with amyotrophic lateral sclerosis treated at home, telemonitoring reduces healthcare utilization and costs [18,58]. Further studies are needed to better assess the clinical benefits, workload, organization, and costs related to telemonitoring.

## 5. Conclusions

Monitoring NIV used overnight at home is a must. There are several tools currently available that vary in terms of level of complexity for the patient and the interpretation of data, as well as in their cost (Figure 11). Basic monitoring combines the symptoms, daytime arterial blood gas exchange, interpretation of data recorded by the ventilator, and regular measurements of nocturnal gas exchange using nocturnal oximetry or transcutaneous capnography. This basic package provides detailed information about the daily use of the treatment and unintentional leaks, as well as the persistence of nocturnal hypoventilation. It also gives some indication of upper airway obstructions and patient–ventilator asynchronies. However, to better characterize these conditions, there is sometimes a need for advanced monitoring tools such as respiratory polygraphy and a detailed interpretation of the pressure, flow, and leak waveforms. Telemonitoring of ventilators should be integrated into this strategy, as it has the potential to improve the quality of NIV and the efficiency of the medical service.

## Figures and Tables

**Figure 1 jcm-12-02163-f001:**
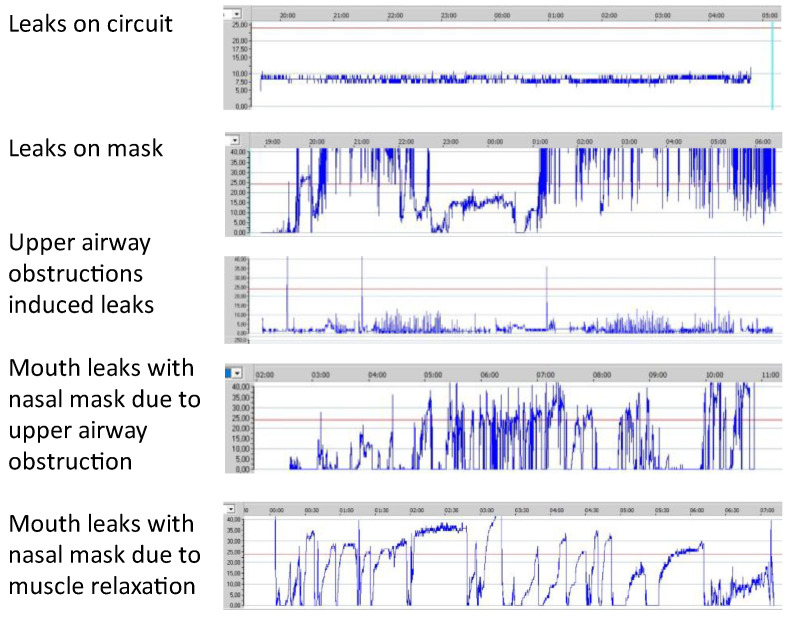
Typical profiles of unintentional leaks repartition overnight.

**Figure 2 jcm-12-02163-f002:**
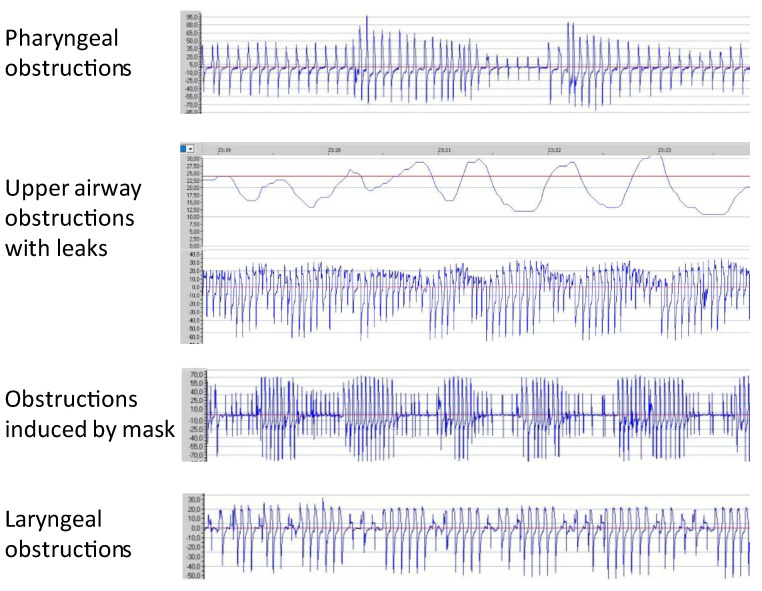
The flow waveform examination may show some respiratory events that are not always detected by the AHI. The shape of the decrease in flow suggests some mechanism and localization of the obstruction. However, a respiratory polygraphy is needed to characterize the mechanism.

**Figure 3 jcm-12-02163-f003:**
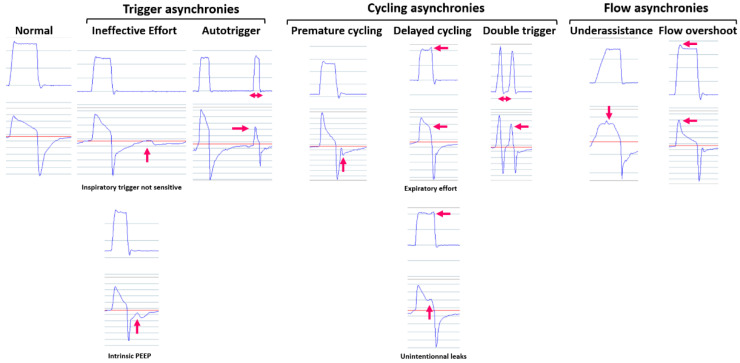
The most common patient–ventilator asynchronies can be detected by monitoring detailed pressure and flow waveforms. Red arrows show where is the asynchrony.

**Figure 4 jcm-12-02163-f004:**
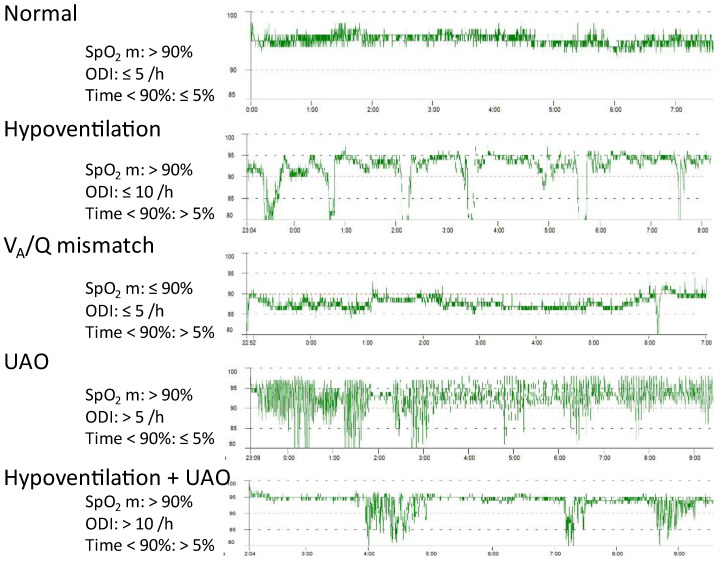
Nocturnal pulse oximetry measurement using a stand-alone device in patients using NIV. Typical results in statistics and on visual inspection of the SpO_2_ curve for different types of respiratory events. Mechanisms of desaturation may coexist, which affect the SpO_2_ curve. SpO_2_ m: mean SpO_2_; ODI: oxygen desaturation index; V_A_/Q: ventilation perfusion ratio; UAO: upper airway obstructions.

**Figure 5 jcm-12-02163-f005:**
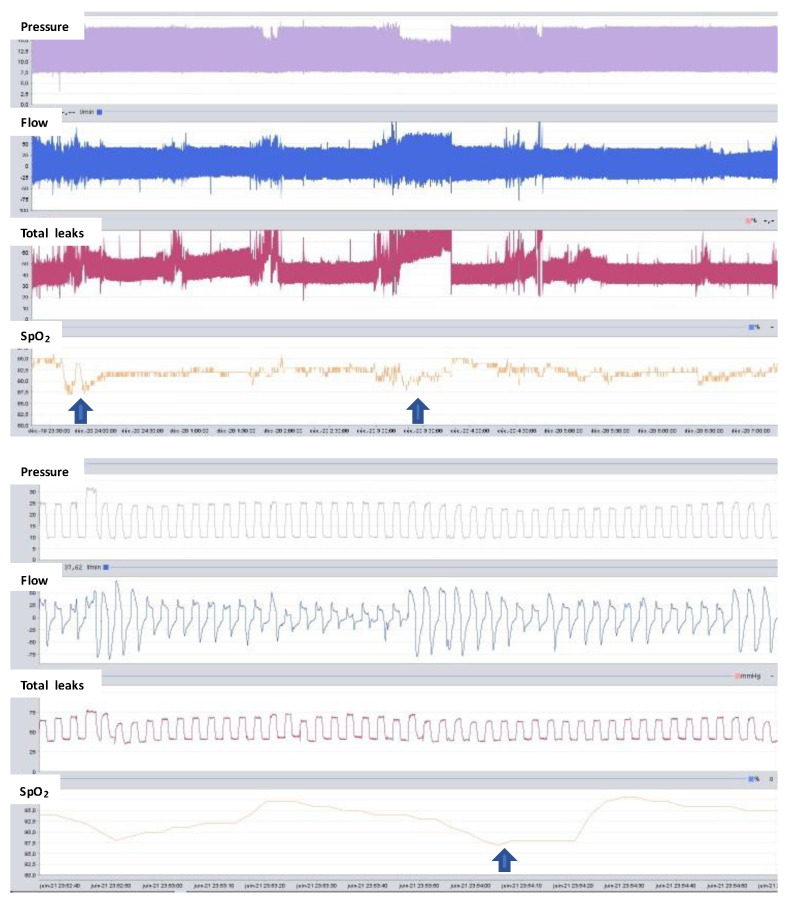
Nocturnal pulse oximetry using a sensor connected to the ventilator. Upper panel shows the full night recording of pressure, flow, total leaks, and SpO_2_. Two prolonged desaturations marked with the arrow occur due to large increase in leakages. Lower panel show a focus over a 2 min period. A short desaturation marked with the arrow occurs, following a decrease in flow without an increase in leaks, suggesting an upper airway obstruction.

**Figure 6 jcm-12-02163-f006:**
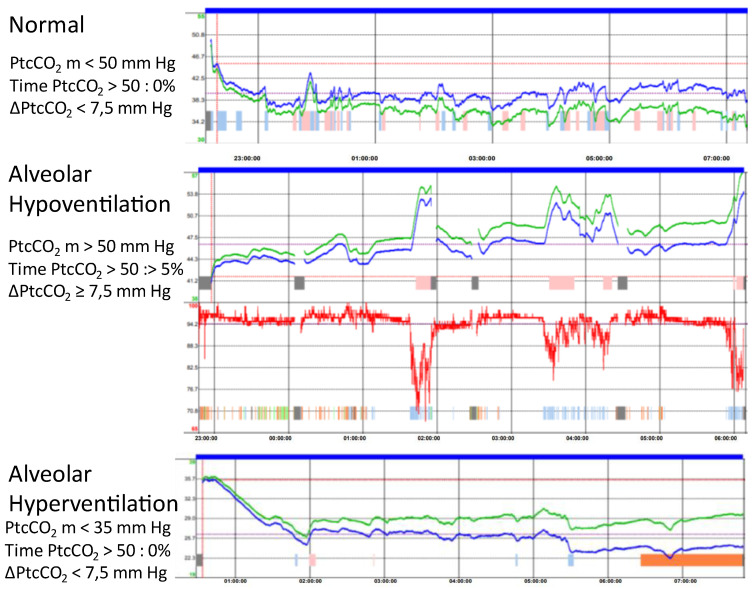
Transcutaneous capnography. Typical results in statistics and on visual inspection of the PtcCO_2_ curve for different types of respiratory events. ΔPtcCO_2_: difference between the maximum and the baseline PtcCO_2_. Green line is raw measurement. Blue line is measurement corrected for drift.

**Figure 7 jcm-12-02163-f007:**
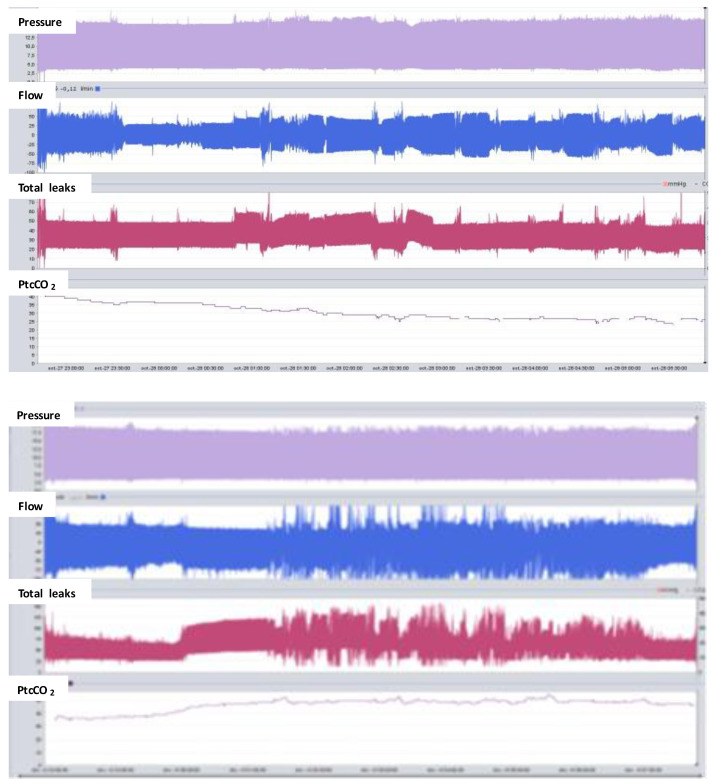
Nocturnal transcutaneous capnography using a device connected to the ventilator. Upper panel shows the full night recording of pressure, flow, total leaks, and raw measurement of PtcCO_2_. PtcCO_2_ decreases along the night from 40 to 30 mmHg, suggesting hyperventilation. Lower panel shows an increased PtcCO_2_ along the night from 40 to 52 mmHg, suggesting alveolar hypoventilation. However, there is a concomitant increase in total leaks, which means that nonintentional leaks are the main cause of PtcCO_2_ increase.

**Figure 8 jcm-12-02163-f008:**
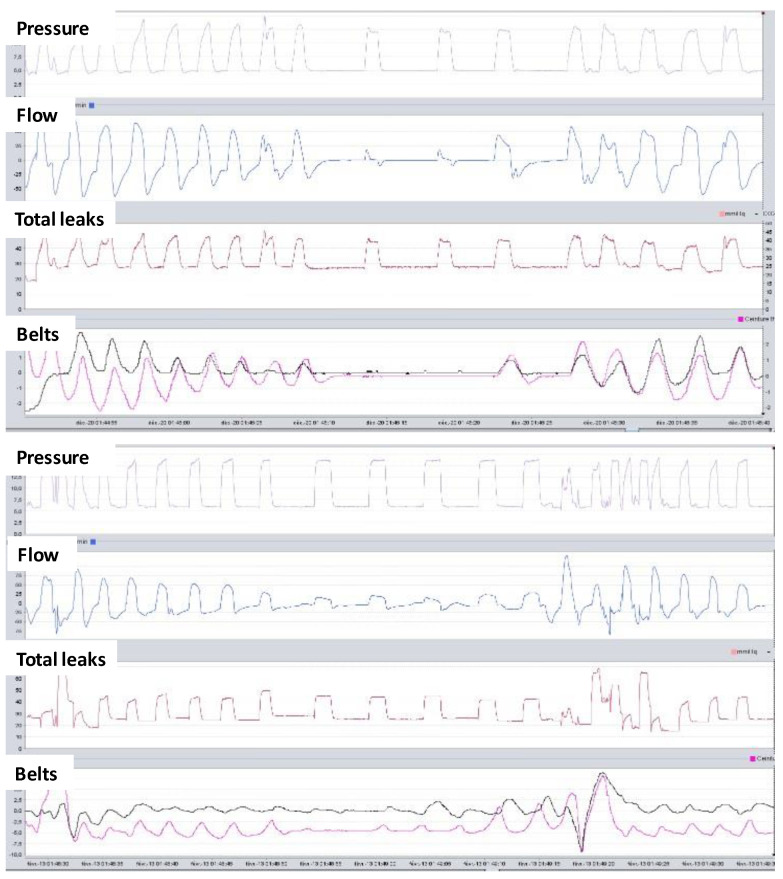
Polygraphy combined with the ventilator. Upper panel shows an upper airway obstruction without unintentional leaks. Belts do not move during the event, suggesting a laryngeal obstruction due to a decrease in the respiratory drive. Lower panel also shows an upper airway obstruction without unintentional leaks. Belts show a phase opposition with a negative inflexion on the thoracic belt (pink) and a positive inflexion of the abdominal belt (black) during the event. This feature suggests a pharyngeal airway obstruction with preserved respiratory drive.

**Figure 9 jcm-12-02163-f009:**
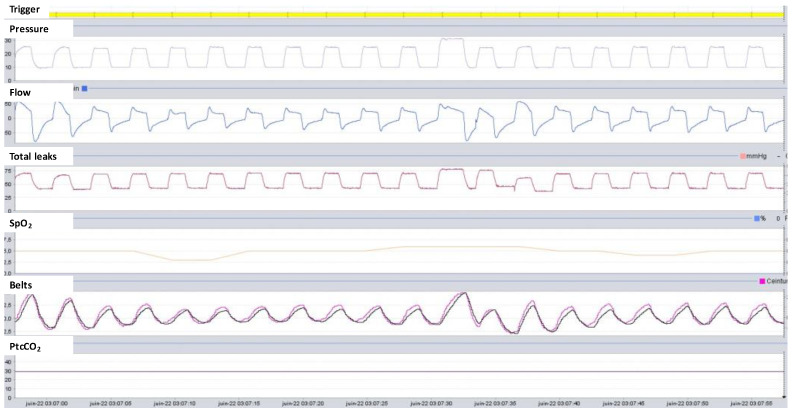
Example of respiratory polygraphy when the belts are connected directly to the ventilator. The ventilator built-in software display from top to bottom: breath trigger color code, pressure, flow, total leaks, SpO_2_, effort belts (thoracic belt in pink and abdominal belt in black), and transcutaneous capnography measured by an external device connected to the ventilator.

**Figure 10 jcm-12-02163-f010:**
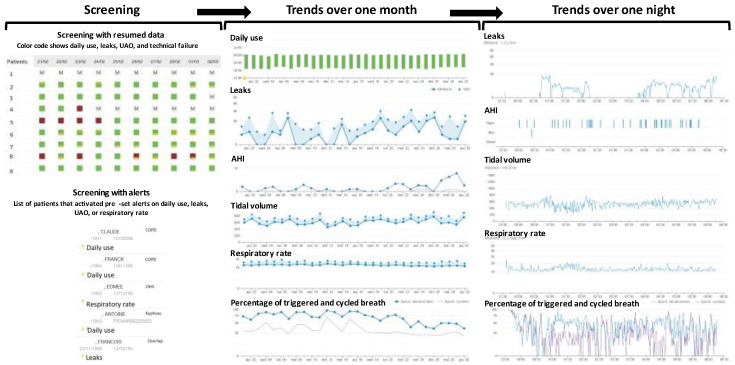
Telemonitoring of NIV. The workflow starts by screening the patients with abnormalities and focuses on trends over one month and one night. Green and red square inform about the daily use above or below 5 h per day, respectively. AHI: Apnea Hypopnea Index.

**Figure 11 jcm-12-02163-f011:**
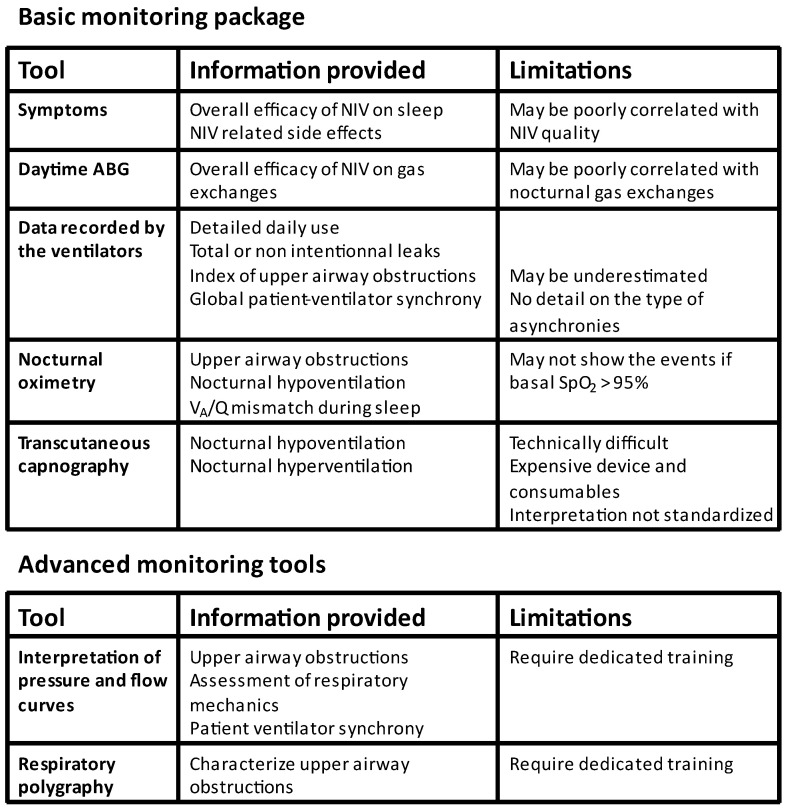
Strategy to monitor NIV. The basic monitoring package is recommended for all patients while advanced monitoring tools are only required in selected patients.

**Table 1 jcm-12-02163-t001:** Accuracy of the variables recorded by the ventilators to be used as clinical information.

Accuracy	Variables	Comments
Good	Mean/Median Daily use	
	Hours of use	
	Total leaks	
	Respiratory rate	Ventilatory rate ≠ patient’s respiratory rate
	Percentage of triggered breath	
	Percentage of spontaneous cycled breath	
	Inspiratory time	
	I/E ratio	
Acceptable	Unintentional leaks	Better if intentional leaks are adjustable
	Apnea hypopnea index	Depends on the type of obstruction and the presence of leaks
Sub-optimal	Tidal volume	±20% if no leaks, less precise if additional unintentional leaks
	Minute volume	

## Data Availability

Not applicable.

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
