# Peer review of "Monitoring Systems in Home Ventilation"

_jcm, 2023, doi:10.3390/jcm12062163_

Round 1
Reviewer 1 Report
I am grateful for the chance to revise this review entitled, "Monitoring Systems in Home Ventilation." Arnal, Oranger, and Gonzalez-Bermejo elegantly explain the monitoring of NIV used at home for an adult population, with a focus on the respective values and limitations of data recorded by the ventilator, the measurement of gas exchanges by nocturnal oximetry and transcutaneous capnography, and patient effort. Furthermore, they detail the possibilities offered by telemonitoring solutions. The bibliographic references are proper, and the English is good. The concepts described are accurate.
However, I would just like to make a minor comment:
Table 4: Why do the authors claim that in a nocturnal hypoventilation syndrome (without UAO), the ODI should be ≤5/h? What classification would the authors choose for a patient with no chronic respiratory disease and a respiratory poligraphy with IAH <5, but just an ODI >5/h and a CT90 >5%? In other words, if that is not hypoventilation, how could the authors classify a patient with a nocturnal oximetry with these results: "SpO2m: >90%, Time <90%: >5%, and ODI >5 /h in a patient without an obstructive sleep disorder (AHI <5)?" In my opinion, this graphic (displayed as it is) may lead to misunderstandings: for example, an hypoventilation syndrome could also coexist with a V/Q mismatch as in scoliosis or chest wall deformities. Also, if the patient does not have a strong predictive value for having a specific disease, I think that nocturnal oximetry is a valuable but extremely simplistic study to perform. Please explain or describe these facts better in the text and in the graphic to avoid wrong assumptions or confusion to the readers.
Still, I would like to congratulate the authors for this review, since they were able to explain complex facts in a simple and conceptual way.
Author Response
We would like to thank Reviewer 1 for the comments that helped us to improve and clarify the manuscript.
Table 4: Why do the authors claim that in a nocturnal hypoventilation syndrome (without UAO), the ODI should be ≤5/h? What classification would the authors choose for a patient with no chronic respiratory disease and a respiratory poligraphy with IAH <5, but just an ODI >5/h and a CT90 >5%? In other words, if that is not hypoventilation, how could the authors classify a patient with a nocturnal oximetry with these results: "SpO2m: >90%, Time <90%: >5%, and ODI >5 /h in a patient without an obstructive sleep disorder (AHI <5)?"
Thank you with this comment. After reviewing our cases, we do agree with the reviewer that in case of hypoventilation syndrome without UAO, ODI is often above 5 but always below 10. Therefore, we have changed the figure 4 accordingly.
In my opinion, this graphic (displayed as it is) may lead to misunderstandings: for example, an hypoventilation syndrome could also coexist with a V/Q mismatch as in scoliosis or chest wall deformities.
We do agree with this comment. What figure 4 want to show are typical shapes of nocturnal SpO2 in patients using NIV for the different causes of desaturation. Of course, mechanism can coexist and the shape of SpO2 will be affected. This comment was added in the legend of figure 4: Mechanism of desaturation may coexist which affect the SpO2 curve.
Also, if the patient does not have a strong predictive value for having a specific disease, I think that nocturnal oximetry is a valuable but extremely simplistic study to perform. Please explain or describe these facts better in the text and in the graphic to avoid wrong assumptions or confusion to the readers.
We do agree with this comment. We believe that nocturnal oximetry for screening nocturnal event in patients with low pre-test probability of having a specific disease has poor sensibility and specificity. However, in this article, we describe nocturnal oximetry as a monitoring tool for patient using NIV at night. These patients have a high pre-test probability of having UAO or hypoventilation. This comment was added in the manuscript: Sensibility and specificity are not optimal when pre-test probability of having specific disease is low. Therefore, clinicians should use it with caution as screening tool. Conversely, in patients using NIV at night, the pre-test probability of having upper airway obstruction and/or nocturnal hypoventilation is high, which makes nocturnal pulse oximetry a valuable tool.
Also, it was précised in the legend of figure 4 that we describe nocturnal oximetry for patients using NIV: Nocturnal pulse oximetry measurement using a stand-alone device in patients using NIV.
Thank you for the reviewing.
Reviewer 2 Report
The manuscript is well-documented on monitoring systems in home ventilation comprehensively. The authors describe the topic in detail, and the manuscript is well reviewed, integrating previous research. There does not appear to be anything particularly amended about the content of the manuscript itself. I present the following opinions on this manuscript so that the authors can read it easily and have interest in the manuscript.
2. Data recorded by the ventilator and 4. Telemonitoring: As readers read, each paragraph seems to have a theme. There are no subheadings, so readers may miss the topic or find it difficult to read. Please add a subheading.
Figures: There are many figures. The authors mainly showed the graphs being monitored, but what if the authors showed the device itself or the patient with the device? In the Telemonitoring section, it would be nice to show the platform implemented in the web and the variables monitored in the platform. Recently, there is also a near real-time display through the ventilator's LTE.
Typo: Figure 10. Oxymetry ---> oximetry
Thank you for your contribution.
Author Response
We would like to thank the reviewer for the suggestions and positive feedback.
- Data recorded by the ventilator and 4. Telemonitoring: As readers read, each paragraph seems to have a theme. There are no subheadings, so readers may miss the topic or find it difficult to read. Please add a subheading.
Thank you for this suggestion. We have added subheadings.
Figures: There are many figures. The authors mainly showed the graphs being monitored, but what if the authors showed the device itself or the patient with the device?
Thank you for this suggestion. However, we consider that there are already a lot of figures and one picture of patient would add very little information.
In the Telemonitoring section, it would be nice to show the platform implemented in the web and the variables monitored in the platform. Recently, there is also a near real-time display through the ventilator's LTE.
Thank you for this suggestion. We have added a figure to show what information we can monitor by telemonitoring and the workflow.
Typo: Figure 10. Oxymetry ---> oximetry
Done
Thank you for the reviewing.
Reviewer 3 Report
Dear authors,
Thank you for your effort on this manuscirpt.
I am satisfied with the manuscript.
Thank you.
Author Response
Thank you for your review and positive feedback